# Facile access to C-glycosyl amino acids and peptides via Ni-catalyzed reductive hydroglycosylation of alkynes

Yan-Hua Liu [1], Yu-Nong Xia [1], Tayyab Gulzar[1], Bingcheng Wei[1], Haotian Li[1], Dapeng Zhu[1], Zhifei Hu[1], Peng Xu[1]✉ & Biao Yu [1,2]✉

C-Glycosyl peptides/proteins are metabolically stable mimics of the native glycopeptides/proteins bearing O/N-glycosidic linkages, and are thus of great therapeutic potential. Herein, we disclose a protocol for the syntheses of vinyl C-glycosyl amino acids and peptides, employing a nickel-catalyzed reductive hydroglycosylation reaction of alkyne derivatives of amino acids and peptides with common glycosyl bromides. It accommodates a wide scope of the coupling partners, including complex oligosaccharide and peptide substrates. The resultant vinyl C-glycosyl amino acids and peptides, which bear common O/N-protecting groups, are amenable to further transformations, including elongation of the peptide and saccharide chains.

[1] State Key Laboratory of Bioorganic and Natural Products Chemistry, Center for Excellence in Molecular Synthesis, Shanghai Institute of Organic Chemistry, University of Chinese Academy of Sciences, Chinese Academy of Sciences, Shanghai, China. [2] School of Chemistry and Materials Science, Hangzhou Institute for Advanced Study, University of Chinese Academy of Sciences, Hangzhou, China. ✉email: peterxu@sioc.ac.cn; byu@sioc.ac.cn

Glycosylation of proteins, involving conjugation of saccharides onto the amino acid residues of proteins, represents a ubiquitous type of posttranslational modification. The added saccharides can then modulate the properties and functions of the proteins in various biological processes, such as in cell adhesion, signal transduction, and immune response[1–5]. In nature, more than 13 monosaccharides can join with eight amino acid residues to provide at least 41 distinct types of glycosidic linkages connecting the saccharides with the proteins[6]. These linkages are mostly O/N-glycosidic bonds with the hydroxyl and amido groups pending on serine, threonine, or asparagine residues[7–9], with Man-Trp being the only C-glycosidic motif known to date[10, 11] (Fig. 1a). The naturally occurring O/N-glycosidic linkages are metabolically vulnerable thus potentially hamper the therapeutical use of glycopeptide/proteins. Thus, the pursuit of hydrolytically stable linkages (e.g., C- or S-glycosidic bonds) in replace of the O/N glycosidic linkages has elicited great interest in the development of glycopeptide/protein drugs[12–15].

In comparison to the preparation of the native O/N- or artificial S/Se-glycosyl peptides[16–21], the construction of C-glycosyl peptides is much more difficult and has lagged far behind[22–25]. Given the markedly lower nucleophilicity and higher $pK_a$ of C–H compared to the X–H (X = O, N, S) counterparts, the conventional glycosylation involving nucleophilic addition onto sugar oxocarbenium intermediates become frequently futile for C-glycosylation. Besides, the complex functionality of peptides are poorly tolerated with the glycosylation conditions. In recent years, transition metal-catalyzed C-glycosylation has gained great attention[26–43] and a large variety of C-glycoside natural products as well as drug candidates have been successfully synthesized[44–49]. However, synthesis of complex C-glycosyl peptides, especially a convergent synthesis using oligosaccharides as donors still poses a formidable challenge, due to the following methodological limits: (i) scarcity of methods for construction of alkyl/alkenyl C-glycosidic bonds, in contrast to the well-studied aryl C-glycosylation; (ii) harsh reaction conditions, including high temperature, strong bases, stoichiometric amount of

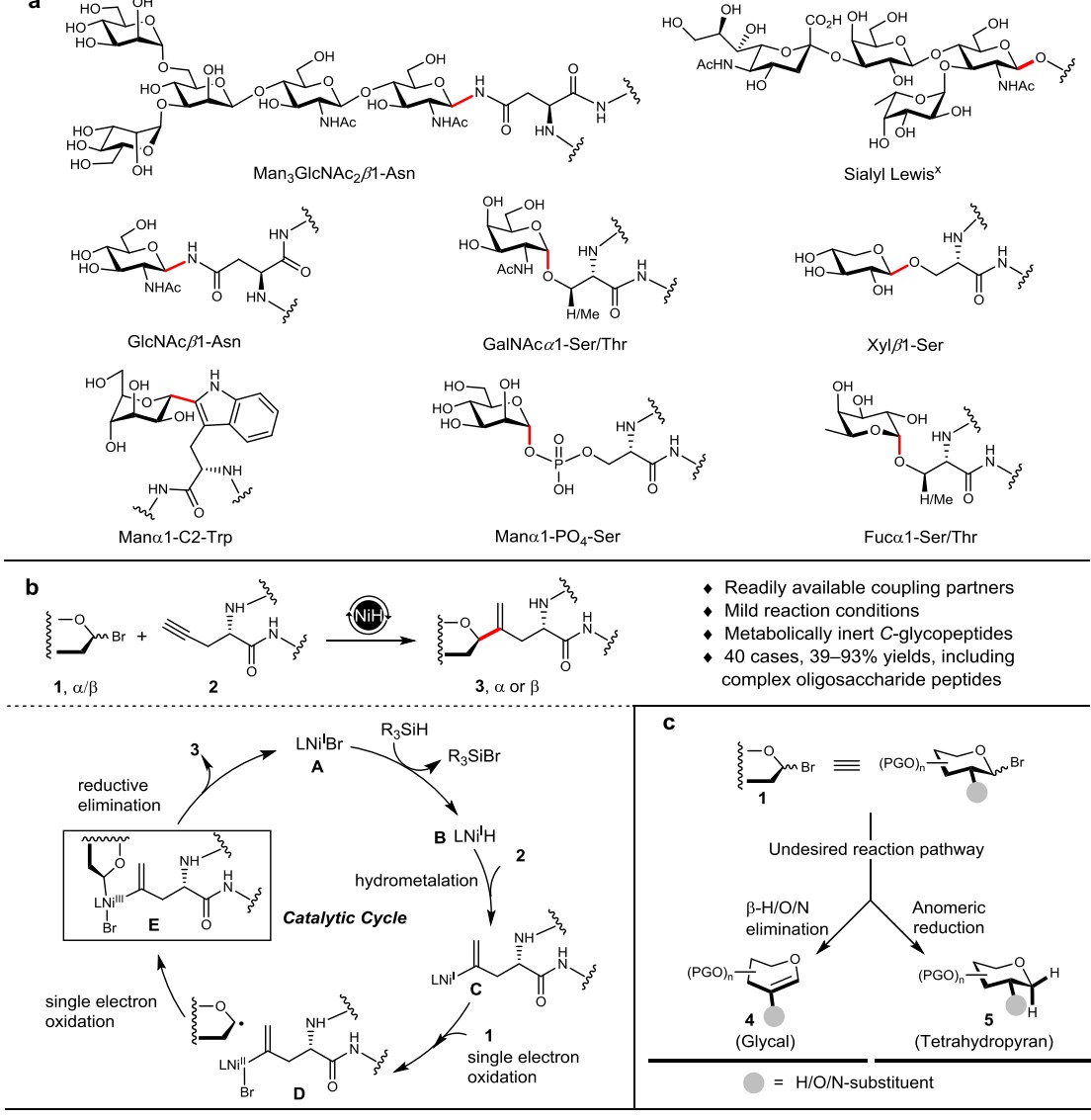

**Fig. 1 Ni-catalyzed syntheses of vinyl C-glycosyl amino acids and peptides. a** The N-linked core Man₃GlcNAc₂β1-Asn motif in glycoproteins, the O-linked tumor-associated carbohydrate antigen Sialyl Lewisˣ in glycolipids, and the native N/O/C-glycosidic linkages in glycoproteins. **b** Nickel-catalyzed reductive hydroglycosylation for access to vinyl C-glycopeptides and a plausible mechanism. **c** Structure of the glycosyl donors **1** and potential by-products **4** and **5**. The glycosidic bonds are highlighted in red.

organometallic reagents, or metal additives that are poorly compatible with peptide substrates; (iii) use of large excess of sugar donors and/or insufficient anomeric selectivity, impeding convergent synthesis with expensive oligosaccharide donors; (iv) use of highly functionalized sugar donors, necessitating multistep transformations to procure the final glycopeptides. Recently, Chen[50], Niu[51], Ackermann[52, 53], Liu[54], Wang[55], and co-workers have disclosed a series of methods for the synthesis of C-glycosyl amino acids via either C–H activation or radical addition strategies. Very recently, Wang et al. reported a stereodivergent synthesis of C-glycosamino acids using glycal donors via Pd/Cu dual catalysis[56]. Notwithstanding, straightforward and practical C-glycosylation methods are still in high demand to conquer the aforementioned limits.

Inspired by the recent breakthrough in the NiH-catalyzed hydrocarbonation of unsaturated bonds[57–61], we envisioned the construction of vinyl C-glycosyl amino acids and peptides via a plausible reaction mechanism as depicted in Fig. 1b. Thus, the catalytic cycle started with a branch-selective insertion of NiH (**B**) to terminal alkyne **2** to form vinyl nickel species **C**, which was then oxidized by glycosyl bromide **1** via a bromine atom abstraction followed by anomeric radical trapping to form high-valent Ni[III] complex **E**. Subsequent reductive elimination delivered the desired C-glycoside **3** and catalyst **A**. The active NiH species **B** was regenerated by hydride transfer from the hydrosilane. It was expected that judicious choice of coupling partners and reaction conditions was required, in order to avoid the β-H/O/N elimination and anomeric reduction that would result in glycal **4** and tetrahydropyran **5** (Fig. 1c), to achieve useful anomeric α/β selectivity, and to secure wide compatibility of the functional groups and protecting groups on the saccharide and peptide substrates.

Here, we show that a wide variety of the easily accessible acetylenic amino acids/peptides and glycosyl bromides can be coupled regio- and stereoselectively under the catalysis of Ni to provide the metabolically stable vinyl C-glycosyl amino acids and peptides.

## Results

**Reaction design and optimization.** To implement the hypothesis, α-mannosyl bromides were initially selected as glycosyl donors and n-hexyne as a model alkyne acceptor. Conditions optimization was proven tedious, and competing by-products from β-H/O elimination (i.e., **4**) or anomeric reduction (i.e., **5**) were obtained concomitantly in many of the cases (see Supplementary Figs. 26–35). Fortunately, extensive surveys of various parameters, including the protecting groups on the sugar bromides, Ni catalyst, bipyridine ligand, phosphine additive, base, silane, reaction atmosphere, and solvent, led to optimal Conditions I and Conditions II for the conjugation of mannose and glucosamine type saccharides (**1a** and **1b**) with N-Boc-L-Pra-OMe (**2a**) (Pra = propargylglycine), yielding vinyl glycosyl amino acids **3aa** and **3ba** in 77 and 85% yield, respectively (Fig. 2). The 1,1-disubstituted alkene moiety in the product is well diagnostic in the [1]H NMR spectra by two singlet signals at high field (e.g., 5.74 and 5.51 ppm for **3aa**; 5.10 and 4.94 ppm for **3ba**)[57]. Besides, the anomeric H of α-glycoside **3aa** presents as a singlet at 4.75 ppm, while the anomeric H of β-glycoside **3ba** is a doublet at 5.06 ppm (d, J = 10.6 Hz). Some key findings with GlcNPhth bromide **1b** as the donor are listed in Fig. 2. Thus, an inert atmosphere was essential for the successful transformation (entries 2 and 3). The absence of dtbbpy ligand completely shut down the reaction (entry 5). Ni(0) could also be used as a catalyst albeit leading to lower yields (entries 6 and 7). The phosphine additive (R)-Tol-BINAP or Ph₃P was found to be fully oxidized into Tol-BINAP (O)₂ or Ph₃P(O) after the reaction, and its absence only slightly diminished the coupling yield (entry 4). Besides, the chirality of Tol-BINAP did not affect the β-selectivity of the glycosylation (see Supplementary Figs. 30, 31, and 35). Therefore, the

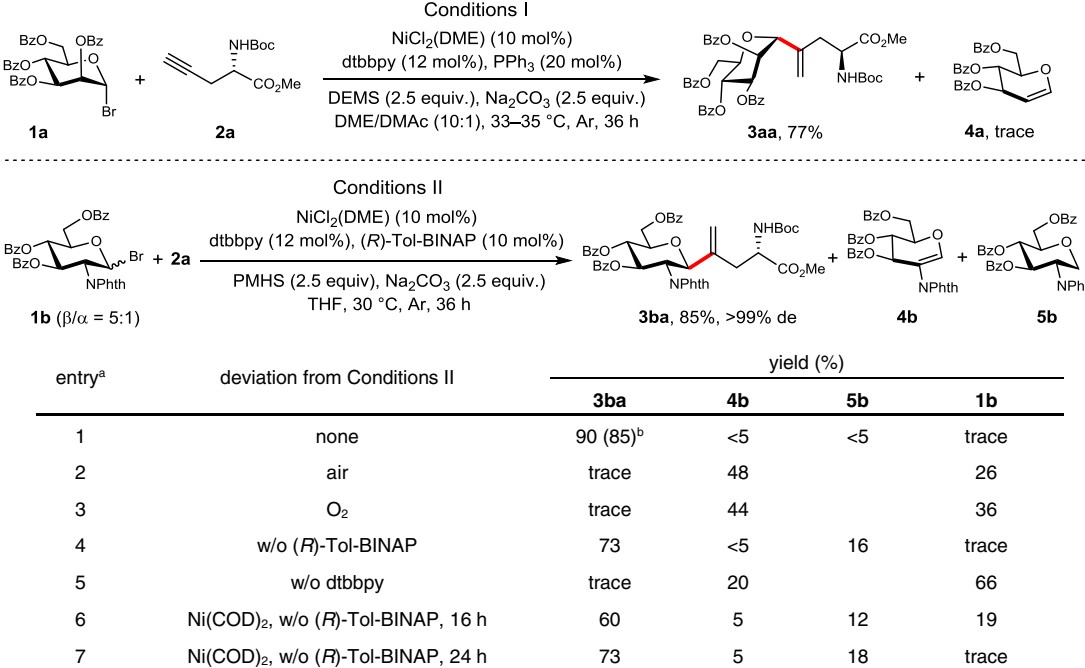

| entry[a] | deviation from Conditions II | yield (%) | | | |
|---|---|---|---|---|---|
| | | **3ba** | **4b** | **5b** | **1b** |
| 1 | none | 90 (85)[b] | <5 | <5 | trace |
| 2 | air | trace | 48 | | 26 |
| 3 | O₂ | trace | 44 | | 36 |
| 4 | w/o (R)-Tol-BINAP | 73 | <5 | 16 | trace |
| 5 | w/o dtbbpy | trace | 20 | | 66 |
| 6 | Ni(COD)₂, w/o (R)-Tol-BINAP, 16 h | 60 | 5 | 12 | 19 |
| 7 | Ni(COD)₂, w/o (R)-Tol-BINAP, 24 h | 73 | 5 | 18 | trace |

**Fig. 2 Optimized reaction conditions I and II and the control experiments for Conditions II.** a Reaction conditions: **1b** (0.1 mmol), **2a** (2.0 equiv.), NiCl₂(DME) (10 mol%), dtbbpy (12 mol%), (R)-Tol-BINAP (10 mol%), PMHS (2.5 equiv.), Na₂CO₃ (2.5 equiv.), THF (1 mL), 30 °C, Ar, 36 h. The yields were determined by [1]H NMR using CH₂Br₂ as an internal standard. b Isolated yield. DME dimethoxyethane, DEMS diethoxymethylsilane, DMAc N,N-dimethylacetamide, dtbbpy 4,4′-di-tert-butyl-2,2′-bipyridine, PMHS poly(methylhydrosiloxane), Tol-BINAP 2,2′-bis(di-p-tolylphosphino)-1,1′-binaphthyl, w/o without. In red are the formed C-glycosidic bonds.

**Fig. 3 C-Mannosylation of alkyne derivatives of amino acids and peptides.** **1a** (0.1 mmol), **2** (2.0 equiv.), NiCl$_2$(DME) (10 mol%), dtbbpy (12 mol%), PPh$_3$ (20 mol%), DEMS (2.5 equiv.), Na$_2$CO$_3$ (2.5 equiv.), DME/DMAc (10/1, v/v, 1 mL), 25–35 °C, Ar, 36 h. Isolated yields are reported. In red are the formed C-glycosidic bonds.

phosphine additive was not involved in the catalytic cycle, whereas it might facilitate the dissolution of NiCl$_2$(DME) and formation of NiCl$_2$(dtbbpy) as the actual catalyst, in addition, a role as a residue O$_2$ scavenger was also possible[62, 63].

It is worth noting that no epimerization of the amino acids was observed in the reaction, as determined by careful HPLC analysis (see Supplementary Figs. 40 and 41), testifying the mild reaction conditions using weak bases (Na$_2$CO$_3$) and mild temperature (<35 °C) for the present C-glycosylation.

**Substrate scope.** With the optimal conditions in hand, we explored the scope of the present method. Firstly, a variety of acetylenic amino acid derivatives, which were easily prepared (see Supplementary Figs. 13–25), were examined to couple with mannosyl bromide **1a** (Fig. 3). Gratifyingly, the frequently used amino protecting groups for peptide synthesis, such as Boc (**3aa**), Cbz (**3ab**), Fmoc (**3ac**) were well tolerated, and so did the

carboxylic acid protecting groups, such as Bn (**3ad**) and $^t$Bu (**3ae**). Expectedly, N-

Phth-L-Pra-OMe reacted smoothly to afford **3af** in 73% yield; alkynes easily derived from natural amino acids via ether or amide linkages, such as propargyl Ser and Tyr ethers (**3ag** and **3ah**), hept-6-ynoyl Lys amide (**3ai**), and propargylamino Asn (**3aj**) were also shown to be suitable substrates. Significantly, the current nickel-catalyzed coupling reaction was highly compatible with the peptide bonds, and thus could be readily applied to the C-glycosylation of dipeptides and tripeptides. Indeed, a panel of the vinyl C-glycosyl dipeptides, including N-Boc-(Man-vinyl)-Ala-Phe-OMe (**3ak**), N-Boc-(Man-vinyl)-Ala-Leu-OMe (**3al**), N-Boc-(Man-vinyl)-Ala-Ile-OMe (**3am**), N-Boc-(Man-vinyl)-Ala-Pro-OMe (**3an**), N-Boc-Trp-(Man-vinyl)-Ala-OMe (**3ao**), O-Bn-N-Boc-Thr-(Man-vinyl)-Ala-OMe (**3ap**), N-Boc-Phe-(Man-vinyl)-Ala-OMe (**3aq**), and tripeptides, including N-Boc-(Man-vinyl)-Ala-Ala-Val-OMe (**3ar**), N-Boc-(Man-vinyl)-Ala-Tle-Ala-OMe (**3as**), and N-Boc-(Man-vinyl)-Ala-Ala-O-$^t$Bu-Thr-OMe (**3at**) were successfully prepared in

52–80% yields. It was noted that no epimerization of the amino acid residues was observed.

GlcNAcβ-Asn represents the most common glycosyl amino acid motif on nuclear and cytoplasmic glycoproteins[6], bringing the synthesis of the glucosamine-based C-glycopeptides an important subject. As shown in Fig. 4, the optimal GlcNPhth bromide donor **1b** could be installed not only to amino acid (**3ba**), but also to dipeptides and tripeptides derivatives with the Pra moiety located either at the terminal (**3bl**, **3bt**, and **3bp**) or at an interior position (**3bu**) in satisfactory yields and exclusive β-selectivity. Notably, the peptide sequence of **3bt** simulates the consensus sequence of Asn-X-Thr/Ser (X can be any amino acid except Pro) in the native N-glycan where GlcNAc can be attached.

Next, we turned to test with other types of pyranosides (Fig. 5). N-Phth-galactosamine bromide was smoothly coupled with Pra **2a**, delivering **3ca** with complete β-selectivity in 79% yield. For xylose, 4-(trifluoromethyl)benzoyl group was used as protecting groups to facilitate separation of the coupling products via silica-gel chromatography, thus (Xyl-vinyl)-

Ala (**3da**) was obtained as β/α anomers with a ratio of 3:1 in 65% yield, in that the β anomer adopted $^4C_1$ conformation ($J_{1,2} = 9.7$ Hz) and the α anomer adopted $^1C_4$ conformation (H1 showed a singlet signal). Mannosamine and rhamnose bromides also reacted smoothly with **2a**, providing **3ea** and **3fa** in 72 and 93% yield, respectively. In addition, the orthogonally protected C-GlcN amino acids **3ga** and **3ha** were obtained in 72 and 58% yield from the corresponding glucosamine donors bearing 6-O-TBDPS and 4-O-Bn groups, respectively. Glucosyl bromide was also tested, and the desired (Glc-vinyl)-Ala (**3ia**) was obtained in 52% yield, albeit without β/α selectivity (β/α = 1:1). Moreover, the scope could be expanded to disaccharide bromide donors, with the fully protected N-Boc-(Galβ(1,3)GlcNβ-vinyl)-Ala-OMe (**3ja**),

N-Boc-(GlcNβ(1,4)GlcNβ-vinyl)-Ala-OMe (**3ka**), N-Boc-(Rhaα(1,2)Rhaα-vinyl)-Ala-OMe (**3la**), and N-Boc-(Fucα(1,3)GlcNβ-vinyl)-Ala-OMe (**3ma**) being prepared in synthetically useful yields (54–68%).

Intriguingly, this method could also be extended to internal acetylenic amino acids. As exemplified in Fig. 6, when

unsymmetrically substituted alkyne **2v** was used as the coupling partner, cis-hydroglycosylation with Man bromide **1a** and GlcN bromide **1b** occurred smoothly, leading to the corresponding regio-isomeric C-glycosides (Man **3av1** and **3av2** and GlcN **3bv1** and **3bv2**, respectively) in moderate yields (52 and 47%) and varied regioselectivity (r.r. = 3:1 and 8:1).

The attained stereoselectivity of the C-glycosylation could be attributed to the predominant conformation of the glycosyl radical intermediate, which is stabilized by the interaction of SOMO of the anomeric unpaired electron with lone pair of the ring oxygen and the σ* of the adjacent C2–O/C2–N bond[64–67]. Thus, a mannose-derived radical adopts preferentially a $^4C_1$ conformation, leading to the 1,2-trans (α-selectivity) product in the C-glycosylation. A glucose-derived radical adopts a flexible $B_{2,5}$ conformation, thus the stereoselectivity of C-glycosylation can be shifted from 1,2-cis (α-selectivity) to 1,2-trans (β-selectivity) by using a bulkier protecting group on C2-OH; and for a glucosamine-derived radical bearing the bulky NPhth group at C2, exclusive 1,2-trans (β-selectivity) product can be attained. Due to the lack of C5 substituent, a xylose-derived radical can adopt both the $B_{2,5}$ conformation and $^1C_4$ conformation, thus resulting in a 1,2-trans (β-selectivity) dominated C-glycosylation.

To probe the occurrence of the glycosyl radical (species D in Fig. 1b), we conducted a radical clock experiment (Fig. 7)[31,35]. Thus, δ-olefinic 1-bromo glucoside **6** and alkyne **2a** were subjected to the standard Conditions II; the desired ring-closure product **7** was isolated in 33% yield with mild diastereoselectivity (d.r. = 3:2). Though not conclusive, this result supports the intermediacy of an anomeric radical species.

**Synthetic utilities**. To demonstrate the potential utilities of the current method, we also examined a scale-up reaction and further transformations of the resulting vinyl C-glycosyl amino acid. Thus, compound **3ba** (1.96 g) was obtained in a 65% yield at a 3.6 mmol scale reaction (Fig. 8a, A; see Supplementary Fig. 37). The transformation of the N-Phth to the native NHAc residue is critical for the synthesis of GlcNAcβ-Asn mimics, fortunately, this was realized selectively via sequential treatment with 80% $N_2H_4 \cdot H_2O$, HOAc, and

**Fig. 4 Scope of C-glycosylation with glucosamine donor 1b. 1b** (0.1 mmol), **2** (2.0 equiv.), NiCl$_2$(DME) (10 mol%), dtbbpy (12 mol%), (R)-Tol-BINAP (10 mol%), PMHS (2.5 equiv.), Na$_2$CO$_3$ (2.5 equiv.), THF (1 mL), 30 °C or as noted, Ar, 36 h. Isolated yields are reported. In red are the formed C-glycosidic bonds.

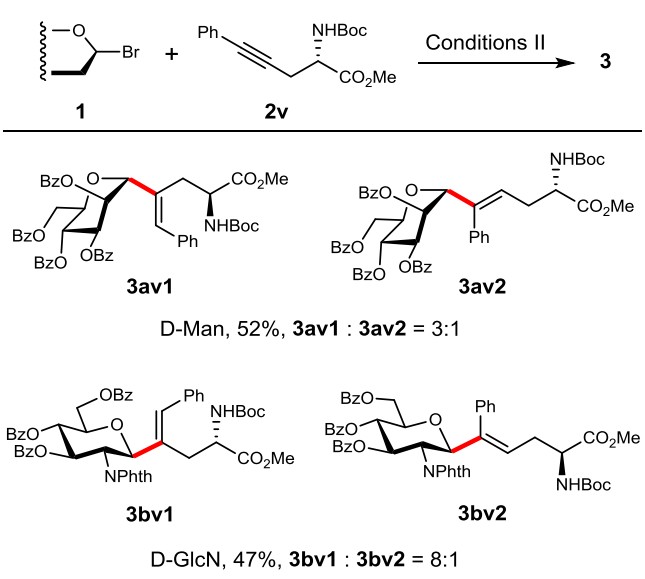

**Fig. 5 C-Glycosylation with various mono- and disaccharide bromides.** See SI for detailed conditions, which might vary slightly from Conditions I and II, and isolated yields are reported. In red are the formed C-glycosidic bonds.

**Fig. 6 C-Glycosylation with an internal acetylenic amino acid.** In red are the formed C-glycosidic bonds.

Ac$_2$O, leading to the desired C-GlcNAc amino acid **8** in 60% yield and >99% de value (see Supplementary Figs. 42 and 43). The orthogonally protected **3ba** and **3ga** allow subsequent elongation of the peptide and saccharide chains. Indeed, the subjection of **3ga** to desilylation followed by fucosylation under the mild Au(I)-catalyzed glycosylation conditions[68] afforded disaccharide **9** in 68% yield (see

Supplementary Fig. 44). Alternatively, the subjection of **3ba** to the cleavage of the N-Boc group followed by peptide synthesis led to C-glycosyl dipeptide **10** in 93% yield (see Supplementary Fig. 45). These transformations showcased the potential of the current protocol for the synthesis of complex and biologically relevant glycopeptides. In addition, three examples of deprotection under strong basic conditions (with LiOH) were conducted, leading to glycosyl amino acids and peptides **11–13** in excellent yields (Fig. 8b).

Finally, we further assessed the feasibility of convergent assembly of C-glycosyl peptides using biologically intriguing oligosaccharides (Fig. 8c). Using a branched pentasaccharide bromide as a donor and Boc-L-Pra-OMe **2a** as acceptor, the desired C-glycosyl amino acid **14** was successfully obtained in ~39% yield (see Supplementary Fig. 36), with the saccharide being relevant to the O-antigen of the lipopolysaccharides of *Pseudomonas syringae*[69]. Using a trisaccharide bromide as a donor, the coupled **15** was obtained in a satisfactory 58% yield, which bears the tumor-associated Lewis[x] antigen[70].

## Discussion

We have developed a nickel-catalyzed hydroglycosylation reaction for the straightforward synthesis of vinyl C-glycosyl amino acids and peptides. A variety of glycosyl bromides can be used as limiting reagents, and excellent 1,2-*trans* diastereoselectivity is attained for C2-axially substituted pyranosides (e.g., Man, ManN, and Rha) or C2-equatorially substituted 2-aminopyranosides (e.g., GlcN and GalN). A wide substrate scope has been proven and also a gram-scale reaction has been demonstrated. The resultant C-glycosyl amino acids and peptides, which bear common N- and O-protecting groups, could be readily transformed into various mimics of the native O/N-glycosyl peptides. The late-

stage *C*-glycosylation with complex oligosaccharide bromides has also been successful. Additionally, the nascent vinyl group in the products would provide a special handle for further derivatization. All these features render the present protocol a promising method for the preparation of *C*-glycosyl peptides of biological and therapeutic significance.

## Methods

**General procedure A (Conditions I) for the NiH-catalyzed reductive hydroglycosylation of acetylenic amino acid and peptides**. To an oven-dried 10 mL Schlenk tube (Titan, TF891910) containing a Teflon coated magnetic stirring bar were added glycosyl bromide **1** (0.1 mmol), NiCl$_2$(DME) (2.2 mg, 10 mol%), dtbbpy (3.2 mg, 12 mol%), PPh$_3$ (5.2 mg, 20 mol%), and Na$_2$CO$_3$ (25 mg, 2.5 equiv.). The tube was sealed with a rubber cap and parafilm, and evacuated then refilled with Ar for at least five cycles. The acetylenic amino acid or peptide derivative **2** was dissolved in solvent (DME/DMAc = 1:1, 1.0 mL) and the solution was injected into the reaction tube (this substrate could be added directly with glycosyl bromide if it was solid). When stirring, (EtO)$_2$MeSiH (40 µL, 2.5 equiv.) was injected via a microliter syringe. Otherwise noted, the tube was moved to an oil bath preheated to 33–35 °C and kept stirring for 36 h. The reaction mixture was diluted with CH$_2$Cl$_2$ (20 mL) and filtered. After concentration, the residue was purified by column chromatography on silica gel or preparative TLC to afford the desired product **3**.

**Fig. 7 A radical clock experiment.** In red are the formed bonds.

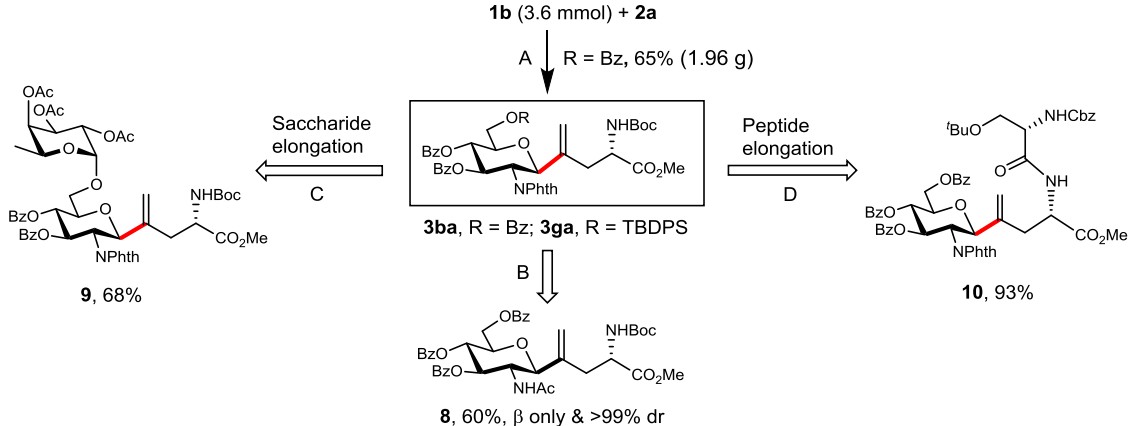

**a. A gram-scale synthesis & subsequent transformations**

**b. Deprotection of acyl groups under basic conditions**

**c. Convergent *C*-glycosylation of complex saccharides with amino acid**

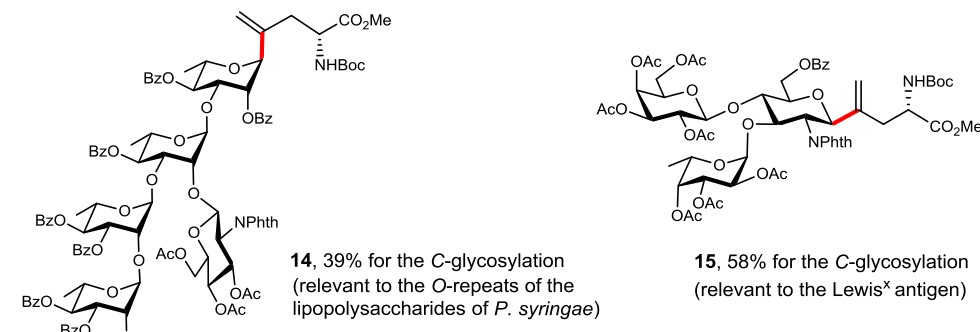

**14**, 39% for the *C*-glycosylation (relevant to the *O*-repeats of the lipopolysaccharides of *P. syringae*)

**15**, 58% for the *C*-glycosylation (relevant to the Lewis$^x$ antigen)

**Fig. 8 Scale-up reaction, subsequent transformation, and *C*-glycosylation with complex saccharides. a** A gram-scale synthesis of *C*-glycosyl amino acid and subsequent transformations. Conditions and reagents: A. NiCl$_2$(DME) (10 mol%), dtbbpy (15 mol%), (*R*)-Tol-BINAP (6.0 mol%), PMHS (2.5 equiv.), Na$_2$CO$_3$ (2.5 equiv.), THF (0.1 M), 25–28 °C, Ar, 48 h, 65%. B. i) 80% N$_2$H$_4$·H$_2$O, MeOH, 0 °C, 9 h; ii) HOAc/MeOH (1/4, v/v), 70 °C, 1.5 h; iii) Ac$_2$O, Et$_3$N, CH$_2$Cl$_2$, 6 h, 60% over three steps. C. i) HF·pyridine, pyridine, 0 °C→rt, 2 h, 85%; ii) Au(PPh$_3$)NTf$_2$ (10 mol%), 4 Å MS, CH$_2$Cl$_2$, 0 °C→rt, 0.5 h, 81%. D. i) CH$_2$Cl$_2$/TFA (2/1, v/v), 0 °C→rt, 1.5 h; ii) *N*-Cbz-*O*-$^t$Bu-L-serine (1.5 equiv.), HOBt (1.5 equiv.), DIPEA (4.0 equiv.), EDCI (1.5 equiv.), DMF, -10 °C→rt, 6 h, 93% over two steps. **b** Deprotection of acyl groups. Conditions: LiOH (7.5 equiv.), MeOH/H$_2$O (4/1, v/v, 0.01 M), rt, 10 h. **c** Convergent *C*-glycosylation of complex oligosaccharides with amino acid, see SI for details. In red are the formed *C*-glycosidic bonds.

**General procedure B (Conditions II) for the NiH-catalyzed reductive hydroglycosylation of acetylenic amino acid and peptides**. To an oven-dried 10 mL Schlenk tube (Titan, TF891910) containing a Teflon coated magnetic stirring bar were added glycosyl bromide **1** (0.1 mmol), NiCl$_2$(DME) (2.2 mg, 10 mol%), dtbbpy (3.2 mg, 12 mol%), (R)-Tol-BINAP (6.7 mg, 10 mol%), and Na$_2$CO$_3$ (25 mg, 2.5 equiv.). The tube was sealed with a rubber cap and parafilm, and evacuated then refilled with Ar for at least five cycles. The acetylenic amino acid or peptide derivative **2** was dissolved in THF (1.0 mL), and the solution was injected into the reaction tube (this substrate could be added directly with glycosyl bromide if it was solid). When stirring, PMHS (32 μL, 2.5 equiv.) was injected via a microliter syringe. Otherwise noted, the tube was kept stirring under an indicated temperature of 30 °C for 36 h. The reaction mixture was diluted with CH$_2$Cl$_2$ (20 mL) and filtered. After concentration, the residue was purified by flash column chromatography on silica gel or preparative TLC to afford the desired product **3**.

## Data availability

The authors declare that all data supporting the findings of this study are available within the paper and its supplementary information file, including experimental details, characterization data, and $^1$H and $^{13}$C NMR spectra of new compounds. All data are available from the corresponding authors upon reasonable request.

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

## Acknowledgements

Financial support from the National Key Research & Development Program of China (2018YFA0507602), National Natural Science Foundation of China (22031011 and 21621002), Key Research Program of Frontier Sciences of CAS (ZDBS-LY-SLH030), Strategic Priority Research Program of CAS (XDB20020000), Youth Innovation Promotion Association of CAS (2020258), Shanghai Municipal Science and Technology Major Project, and Shanghai Committee of Science and Technology (17JC1405300) are acknowledged.

## Author contributions

B.Y., Y.-H.L., and D.Z. conceived the project. Y.-H.L., Y.-N.X., T.G., B.W., H.L., and Z.H. conducted the synthetic work. Y.-H.L., P.X., and D.Z. conducted the data analysis. B.Y., Y.-H.L., and P.X. wrote the manuscript. All authors discussed the results and commented on the paper.

## Competing interests

The authors declare no competing interests.
