## [Peer Review File · Nature Communications]

Reviewers' Comments:

Reviewer #1:

Remarks to the Author:

In this work, Yu, Xu and coworkers reported a novel Ni-catalyzed hydroglycosylation of alkynes, affording a new class of C-glycosides. This reaction demonstrates a remarkable substrate scope and tolerates a broad array of sugar units. The successful preparation of 12 and 13 is particularly impressive, highlighting the power of this reaction. This method will be widely adopted to make C-glycosides that are of high medicinal value.

This reaction is challenging to accomplish. I was surprised that glycosyl bromides (prone to reduction) could be used as substrates. Moreover, the regioselectivity of this method, which gives 1,1-disubstituted alkenes is noteworthy. Conceivably, the alkene unit can serve as a handle for further derivatization, further adding to the utility of this method. I thus support the publication of this work, but have the following suggestions.

First, as mentioned above, the regioselectivity to form vinyl Nickel species 1 is interesting. The authors may provide a rationale for this hypothesis. Could this be because the hydrometallation has a high radical character?

Second, the authors made a keen observation about the fate of phosphine ligands, and proposed that they may not be involved in the catalytic cycle, but instead help to dissolve Ni or scavenge O₂. However, it's also possible that phosphine ligands may just serve to stabilize Ni-catalyst by prevent it from decomposing (to metal nanoparticles). Lipshultz and Buchwald have suggested such a role of bystander phosphine ligands in their Cu-H catalyzed reactions.

Third, it is interesting the authors didn't mention use of glucose derived bromides. They might be difficult substrates, but deserve mentioning in the scope to better define the limits and scope of this method.

Besides, other minor issues include the following:

"prausible" in page 3 is a typo.

Reference 50 should be updated.

Reviewer #2:

Remarks to the Author:

In this work, Xu, Yu and co-workers reported a new method for the stereoselective synthesis of C-glycosides (C-glycosyl amino acids and peptides) via nickel-catalyzed reductive hydroglycosylation of alkynes. Intriguingly, they demonstrated that highly functionalized radicals (glycosyl radicals) could participate in nickel-catalyzed reductive hydroalkylations. As a new C-glycosylation technology, it has the advantage of using readily available glycosyl bromides and unactivated alkynes as coupling partners. Furthermore, this reaction doesn't require organic zinc/magnesium and are tolerant of peptides and oligosaccharides (even tetra/pentasaccharide substrates), which is not possible with most previous transition metal catalyzed C-glycosylation methods (e.g., Fe: *J. Am. Chem. Soc.* 2017, 139, 10693-10701. Co: Nicolas, L.; Angibaud, P.; Stansfield, I.; Bonnet, P.; Meerpoel, L.; Reymond, S.; Cossy, J., Diastereoselective Metal-Catalyzed Synthesis of C-Aryl and C-Vinyl Glycosides. *Angew. Chem. Int. Ed.* 2012, 51, 11101-11104; Ni: *J. Am. Chem. Soc.* 2014, 136, 17645-17651). The scholarly presentation is well organized and fully supported by the Supporting Information. I think it is suitable for publication in this journal (*Nature Communications*).

Some points are raised for the authors to enhance the quality of this manuscript.

Major Points

1. The reaction exhibits excellent 1,2-trans selectivity. The authors should better spend one paragraph discussing the stereoselectivity of C-glycosylation. The stereoselectivity may be rationalized by analyzing the lowest energy conformations of the glycosyl radicals. Owing to the stabilizing interaction of the SOMO with the σ^* of the adjacent C-OR(C2-OR) bond, (e.g., *Angew. Chem. Int. Ed.* 1984, 23, 896-898; *J. Chem. Soc., Perkin Trans. 2*, 1986, 1453-1459; *J. Am. Chem. Soc.* 2017, 139, 10693-10701.), the preferred conformation for mannose-derived glycosyl radicals is 4C1. Thus, C-glycosylation involving mannosyl radicals is generally alpha-selective, consistent with results shown in Fig 2. However, the preferred conformation for radicals with glucose-configuration is (twisted) B2,5. The stereoselectivity could be shifted from alpha-selective (*Org. Lett.* 2002, 4, 4623-4625.) to beta-selective by using a bulkier protecting group on C2-OH/NH2 (*J. Org. Chem.* 1996, 61, 6442-6445, consistent with results in Fig. 3 and 4). Alternatively, the stereoselectivity could also be understood by invoking a model similar to those proposed for previous Ni-catalyzed C-glycosylation (e.g., *Angew. Chem. Int. Ed.* 2000, 39, 4111-4114; *Sci. China Chem.* 2019, 62, 1492-1496.) .

2. I hope the authors could prove the existence of glycosyl radicals. For example, by using a substrate with an alkene attached at C2-OH (e.g., *J. Am. Chem. Soc.* 2017, 139, 10693-10701; *Angew. Chem. Int. Ed.* 2012, 51, 11101-11104.).

Minor points

1. The Ni-catalyzed hydroalkylation of alkyne may give two regio-isomers. Did the authors observe any anti-Markovnikov products? Please comment on this point.
2. P4, please add one or two sentences describing evidence that support "no epimerization of the amino acid residues was observed."
3. delete parentheses on structures 2 and 3 in Fig 3.
4. Ref 56 is not related to NiH catalyzed reaction.
5. The ¹H-spectra of 1c,1d are a bit squeezed. Please adjust the height of the peaks.
6. It is better to add one section for the structure determination into the Supporting Information. For example, select one compound from each series and describe the key features of their spectra (e.g., 1,1-disubstituted alkene vs. 1,2-disubstituted alkene; alpha vs. beta glycoside).

Reviewer #3:

Remarks to the Author:

The manuscript by Xu, Yu and coworkers presents the development of an innovative method for nickel-catalyzed reductive additions to construct hybrids of C-glycosyl motifs and amino acid derivatives. An optimized nickel (hydride) catalyst allowed for the chemo- and regio-selective hydroalkylation of alkynes derived from amino acids. Thereby innovative new structures could be accessed.

The manuscript was very carefully prepared and is well written.

Given the topical interest in organic electrocatalysis, along with the practical importance of aminations, I recommend publication of this fine manuscript after the following minor revision.

- 1) In the scope studies I do miss information on the selectivity when using more sensitive amino acids, such as tyrosine, lysine, serine, and cys.
- 2) The mechanism is proposed to involve a nickel hydrido species. Is there mechanistic support for this for the new method?
- 3) Is reductive homocoupling of the bromide 1a observed in Figure 2?
- 4) The method seems largely limited to terminal alkynes: Information on the use of internal alkynes would be helpful for the reader (even if not successful).
- 5) Are protecting group-free bromides and amino acids viable substrates?

I. Response to Decision Letter (NCOMMS-21-10854)

We are grateful to the reviewers for their kind comments and helpful suggestions that have helped us to improve our paper. As indicated in the point-by-point responses that follow, we have taken all these comments into account in the revised manuscript. The changes made in the revised manuscript are highlighted in yellow.

Reviewer #1 (Remarks to the Author):

In this work, Yu, Xu and coworkers reported a novel Ni-catalyzed hydroglycosylation of alkynes, affording a new class of C-glycosides. This reaction demonstrates a remarkable substrate scope and tolerates a broad array of sugar units. The successful preparation of 12 and 13 is particularly impressive, highlighting the power of this reaction. This method will be widely adopted to make C-glycosides that are of high medicinal value.

This reaction is challenging to accomplish. I was surprised that glycosyl bromides (prone to reduction) could be used as substrates. Moreover, the regioselectivity of this method, which gives 1,1-disubstituted alkenes is noteworthy. Conceivably, the alkene unit can serve as a handle for further derivatization, further adding to the utility of this method. I thus support the publication of this work, but have the following suggestions.

Response: We are grateful to this reviewer for the encouraging comments.

First, as mentioned above, the regioselectivity to form vinyl Nickel species 1 is interesting. The authors may provide a rationale for this hypothesis. Could this be because the hydrometallation has a high radical character?

Response: Thanks for the comment. Indeed, only 1,1-disubstituted alkene type products are obtained in the present reaction, whereas the corresponding 1,2-disubstituted regio-isomers have not been isolated. We attribute this regioselectivity to the “hydronickelation of terminal alkyne step (from int. B to int. C; Fig. 1b)”, where the hydride is transferred to the terminal carbon of the alkyne due to the less bulkiness and thus faster insertion rate to this carbon. However, it is not known about the radical character of the nickel hydride species (species B), and the resultant vinyl nickel species (int. C) might not possess radical character (a vinyl

radical should be a highly reactive species), which could participate in the cross-coupling step with glycosyl bromide **1** to deliver int. D and E. The high valent species E might undergo a rapid reductive elimination to yield product **3**. This plausible mechanism is depicted in Figure 1b.

Second, the authors made a keen observation about the fate of phosphine ligands, and proposed that they may not be involved in the catalytic cycle, but instead help to dissolve Ni or scavenge O₂. However, it's also possible that phosphine ligands may just serve to stabilize Ni-catalyst by prevent it from decomposing (to metal nanoparticles). Lipshultz and Buchwald have suggested such a role of bystander phosphine ligands in their Cu-H catalyzed reactions.

Response: Thanks for the suggestions. Indeed, Lipshultz, Buchwald, and others have suggested such a bystander role of phosphine ligands in the Cu-H and Ni-H catalyzed reactions [for CuH: a) Lipshutz, B. H.; Servesko, J. M. *Angew. Chem., Int. Ed.* **2003**, *42*, 4789; b) Lipshutz, B. H.; Servesko, J. M.; Taft, B. R. *J. Am. Chem. Soc.* **2004**, *126*, 8352; c) Lipshutz, B. H.; Frieman, B. A. *Angew. Chem., Int. Ed.* **2005**, *44*, 6345; d) Zhu, S.; Buchwald, S. L. *J. Am. Chem. Soc.* **2014**, *136*, 15913; e) Ascic, E.; Buchwald, S. L. *J. Am. Chem. Soc.* **2015**, *137*, 4666; f) Bandar, J. S.; Pirnot, M. T.; Buchwald, S. L. *J. Am. Chem. Soc.* **2015**, *137*, 14812; g) Niu, D.; Buchwald, S. L. *J. Am. Chem. Soc.* **2015**, *137*, 9716; for NiH: h) Lu, X.; Xiao, B.; Liu, L.; Fu, Y. *Chem. Eur. J.* **2016**, *22*, 11161; i) Chen, F.; Xu, X.; He, Y.; Huang, G.; Zhu, S. *Angew. Chem. Int. Ed.* **2020**, *59*, 5398; j) Yin, X.; Chen, B.; Qiu, F.; Wang, X.; Liao, Y.; Wang, M.; Lei, X.; Liao, J. *ACS Catal.* **2020**, *10*, 1954]. However, in the present reaction, it is not likely that the phosphine additive serves to stabilize the metal Ni(0), since all the phosphine is oxidized to phosphine oxide during the reaction. For better understanding, we have added two references in the manuscript [*i.e.*, 62. Lipshutz, B. H. & Frieman, B. A. CuH in a bottle: a convenient reagent for asymmetric hydrosilylations. *Angew. Chem. Int. Ed.* **44**, 6345–6348 (2005). 63. Zhu, S. & Buchwald, S. L. Enantioselective CuH-catalyzed anti-Markovnikov hydroamination of 1,1-disubstituted alkenes *J. Am. Chem. Soc.* **136**, 15913–15916 (2014)].

Third, it is interesting the authors didn't mention use of glucose derived bromides. They might be difficult substrates, but deserve mentioning in the scope to better define the limits and scope of this method.

Response: Thanks for the expert suggestion. We have added a reaction with a glucose derived bromide. The reaction gives the desired glucosyl amino acid derivative **3ia** in 52% yield, albeit with a poor β/α selectivity ($\beta/\alpha = 1:1$; Fig. 4). A paragraph discussing briefly the β/α selectivity of C-glycosylation has been added (see Response to the first comment of reviewer #2).

Besides, other minor issues include the following:

“prausible” in page 3 is a typo.

Response: Thanks. The typo has been corrected.

Reference 50 should be updated.

Response: Thanks. Ref. 50 has been updated.

Reviewer #2 (Remarks to the Author):

In this work, Xu, Yu and co-workers reported a new method for the stereoselective synthesis of C-glycosides (C-glycosyl amino acids and peptides) via nickel-catalyzed reductive hydroglycosylation of alkynes. Intriguingly, they demonstrated that highly functionalized radicals (glycosyl radicals) could participate in nickel-catalyzed reductive hydroalkylations. As a new C-glycosylation technology, it has the advantage of using readily available glycosyl bromides and unactivated alkynes as coupling partners. Furthermore, this reaction doesn't require organic zinc/magnesium and are tolerant of peptides and oligosaccharides (even tetra/pentasaccharide substrates), which is not possible with most previous transition metal catalyzed C-glycosylation methods (e.g., Fe: J. Am. Chem. Soc. 2017, 139, 10693-10701. Co: Nicolas, L.; Angibaud, P.; Stansfield, I.; Bonnet, P.; Meerpoel, L.; Reymond, S.; Cossy, J., Diastereoselective Metal-Catalyzed Synthesis of C-Aryl and C-Vinyl Glycosides. Angew. Chem. Int. Ed. 2012, 51, 11101-11104; Ni: J. Am. Chem. Soc. 2014, 136, 17645-17651). The scholarly presentation is well organized and fully supported by the Supporting Information. I think it is suitable for publication in this journal (Nature Communications). Some points are raised for the authors to enhance the quality of this manuscript.

Response: We are grateful to this reviewer for the encouraging comments.

Major Points

1. The reaction exhibits excellent 1,2-*trans* selectivity. The authors should better spend one paragraph discussing the stereoselectivity of C-glycosylation. The stereoselectivity may be rationalized by analyzing the lowest energy conformations of the glycosyl radicals. Owing to the stabilizing interaction of the SOMO with the σ^* of the adjacent C-OR(C2-OR) bond, (e.g., *Angew. Chem. Int. Ed.* 1984, 23, 896-898; *J. Chem. Soc., Perkin Trans. 2*, 1986, 1453-1459; *J. Am. Chem. Soc.* 2017, 139, 10693-10701.), the preferred conformation for mannose-derived glycosyl radicals is ⁴C₁. Thus, C-glycosylation involving mannosyl radicals is generally alpha-selective, consistent with results shown in Fig 2. However, the preferred conformation for radicals with glucose-configuration is (twisted) B_{2,5}. The stereoselectivity could be shifted from alpha-selective (*Org. Lett.* 2002, 4, 4623-4625.) to beta-selective by using a bulkier protecting group on C2-OH/NH₂ (*J. Org. Chem.* 1996, 61, 6442-6445, consistent with results in Fig. 3 and 4).

Alternatively, the stereoselectivity could also be understood by invoking a model similar to those proposed for previous Ni-catalyzed C-glycosylation (e.g., *Angew. Chem. Int. Ed.* 2000, 39, 4111-4114; *Sci. China Chem.* 2019, 62, 1492-1496.) .

Response: Thanks for the suggestions. We have added one paragraph to discuss briefly the stereoselectivity of C-glycosylation following the suggestion of the reviewer. It reads:

The attained stereoselectivity of the C-glycosylation could be attributed to the predominant conformation of the glycosyl radical intermediate, which is stabilized by the interaction of SOMO of the anomeric unpaired electron with lone pair of the ring oxygen and the σ^* of the adjacent C2-O/C2-N bond⁶⁴⁻⁶⁷. Thus, a mannose-derived radical adopts preferentially a ⁴C₁ conformation, leading to the 1,2-*trans* (α -selectivity) product in the C-glycosylation. A glucose-derived radical adopts a flexible B_{2,5} conformation, thus the stereoselectivity of C-glycosylation could be shifted from 1,2-*cis* (α -selectivity) to 1,2-*trans* (β -selectivity) by using a bulkier protecting group on C2-OH; and for a glucosamine-derived radical bearing the bulky NPhth group at C2, exclusive 1,2-*trans* (β -selectivity) product is attained. Due to the lack of C5 substituent, a xylose-derived radical can adopt both the B_{2,5} conformation and ¹C₄ conformation, thus resulting in a 1,2-*trans* (β -selectivity) dominated C-glycosylation.

Accordingly, references 64-67 are added:

64. Giese, B. The stereoselectivity of intermolecular free radical reactions. *Angew. Chem. Int. Ed.* **28**, 969–980 (1989).
65. Roe, B. A., Boojamra, C. G., Griggs, J. L., & Bertozzi, C. R. Synthesis of β -C-glycosides of *N*-acetylglucosamine via Keck allylation directed by neighboring phthalimide groups. *J. Org. Chem.* **61**, 6442–6445 (1996).
66. Togo, H., Wei, H., Waki, Y. & Yokoyama, M. C-Glycosidation technology with free radical reactions. *Synlett* **7**, 700–717 (1998).
67. Abe, H., Shuto, S. & Matsuda, A. Highly α - and β -selective radical C-glycosylation reactions using a controlling anomeric effect based on the conformational restriction strategy. A study on the conformation–anomeric effect–stereoselectivity relationship in anomeric radical reactions. *J. Am. Chem. Soc.* **123**, 11870–11882 (2001).

2. I hope the authors could prove the existence of glycosyl radicals. For example, by using a substrate with an alkene attached at C2-OH (e.g., *J. Am. Chem. Soc.* **2017**, *139*, 10693-10701; *Angew. Chem. Int. Ed.* **2012**, *51*, 11101-11104.).

Response: Thanks for the suggestion. We have performed a radical clock experiment to support the existence of glycosyl radical. Figure 6 and a paragraph are added.

Fig. 6 A radical clock experiment.

To probe the existence of the glycosyl radical species D (Fig. 1b), we conducted a radical clock experiment (Fig. 6).³⁵ Thus, δ -olefinic 1-bromo glucoside **6** and alkyne **2a** were subjected to the standard **Conditions II**; the desired ring-closure product **7** was isolated in 33% yield with a mild diastereoselectivity (d.r. = 3:2). Though not conclusive, this result support the intermediacy of an anomeric radical species.

Accordingly, reference 35 is added:

35: Adak, L. et al. Synthesis of aryl C-glycosides via iron-catalyzed cross coupling of halosugars: stereoselective anomeric arylation of glycosyl radicals. *J. Am. Chem. Soc.* **139**, 10693–10701 (2017).

Minor points

1. The Ni-catalyzed hydroalkylation of alkyne may give two regio-isomers. Did the authors observe any anti-Markovnikov products? Please comment on this point.

Response: Thanks for the suggestion. We didn't observe the anti-Markovnikov products. We attribute this to the hydronicelation of terminal alkyne step (from int. B to int. C), where the hydride was transferred to the terminal carbon position due to the less bulkiness of and thus faster insertion rate to this carbon atom.

2. P4, please add one or two sentences describing evidence that support "no epimerization of the amino acid residues was observed."

Response: Thanks for the suggestion. We have added one sentence, it reads:

It is worth noting that no epimerization of the amino acids was observed in the reaction, as determined by careful HPLC analysis (see Supplementary 2.7), testifying the mild reaction conditions using weak bases (Na_2CO_3) and mild temperature ($<35\text{ }^\circ\text{C}$) for the present C-glycosylation.

3. delete parentheses on structures 2 and 3 in Fig 3.

Response: Thanks. These have been deleted.

4. Ref 56 is not related to NiH catalyzed reaction.

Response: Thanks. Ref. 56 has been deleted.

5. The ^1H -spectra of 1c,1d are a bit squeezed. Please adjust the height of the peaks.

Response: Thanks for the suggestion. These spectra have been adjusted.

6. It is better to add one section for the structure determination into the Supporting Information. For example, select one compound from each series and describe the key features of their spectra (e.g., 1,1-disubstituted alkene vs. 1,2-disubstituted alkene; alpha vs. beta glycoside).

Response: Thanks for the suggestion. We have added a sentence in the context (in page 2), it reads: The 1,1-disubstituted alkene moiety in the product are well diagnostic in the ^1H NMR

spectra by two singlet signals at high field (e.g., 5.74 & 5.51 ppm for **3aa**; 5.10 & 4.94 ppm for **3ba**).⁵⁷ Besides, the anomeric H of α -glycoside **3aa** presents as a singlet at 4.75 ppm, while the anomeric H of β -glycoside **3ba** is a doublet at 5.06 ppm (d, $J = 10.6$ Hz).

In addition, the vinyl H & anomeric H have been highlighted in bold in the revised Supplementary Information.

Reviewer #3 (Remarks to the Author):

The manuscript by Xu, Yu and coworkers presents the development of an innovative method for nickel-catalyzed reductive additions to construct hybrids of C-glycosyl motifs and amino acid derivatives. An optimized nickel (hydride) catalyst allowed for the chemo- and regio-selective hydroalkylation of alkynes derived from amino acids. Thereby innovative new structures could be accessed.

The manuscript was very carefully prepared and is well written.

Given the topical interest in organic electrocatalysis, along with the practical importance of aminations, I recommend publication of this fine manuscript after the following minor revision.

Response: We are grateful to this reviewer for the encouraging comments.

1) In the scope studies I do miss information on the selectivity when using more sensitive amino acids, such as tyrosine, lysine, serine, and cys.

Response: We have included substrates containing Tyr, Lys, and Ser (i.e., **3ah**, **3ai**, and **3ag**). The corresponding reactions proceeded well (Fig. 2).

2) The mechanism is proposed to involve a nickel hydrido species. Is there mechanistic support for this for the new method?

Response: The nickel hydride species is a commonly proposed intermediate in the nickel-silane-catalyzed hydrofunctionalization reactions (see Ref. 57-61), unfortunately, it has never been characterized.

3) Is reductive homocoupling of the bromide **1a** observed in Figure 2?

Response: The reductive homocoupling of bromide **1a**, which would give 1,1-linked C-

disaccharides, has not been observed in the reaction.

4) The method seems largely limited to terminal alkynes: Information on the use of internal alkynes would be helpful for the reader (even if not successful).

Response: Thanks for the suggestion. We have added two experiments with an internal alkyne (*i.e.*, **2v**), and the results are given in Fig. 5.

Fig. 5 C-Glycosylation with an internal acetylenic amino acid.

A paragraph is added, it reads: Intriguingly, this method could also be extended to internal acetylenic amino acids. As exemplified in Fig. 5, when unsymmetrically substituted alkyne **2v** was used as the coupling partner, *cis*-hydroglycosylation with Man bromide **1a** and GlcN bromide **1b** occurred smoothly, leading to the corresponding regio-isomeric C-glycosides (Man **3av1** & **3av2** and GlcN **3bv1** & **3bv2**, respectively) in moderate yields (52% and 47%) and varied regioselectivity (r.r. = 3:1 and 8:1).

5) Are protecting group-free bromides and amino acids viable substrates?

Response: We have not tried protecting group-free glycosyl bromides and amino acids, because of the poor solubility of these substrates.

II. Other revisions

- 1) A “Data Availability” section has been added before the References.
- 2) Two more grants are added in the Acknowledgements, those are “Youth Innovation

Promotion Association of CAS (2020258) and Shanghai Municipal Science and Technology Major Project”.

Reviewers' Comments:

Reviewer #1:

Remarks to the Author:

The authors did a remarkable job on revising this work. All of the concerns raised by reviewers have been satisfactorily addressed. I now support its publication.

Reviewer #2:

Remarks to the Author:

The authors have appropriately addressed all the comments from the reviewers. I am supportive of the acceptance.

Reviewer #3:

Remarks to the Author:

The revised manuscript by Xu, Yu and coworkers has addressed all comments by the reviewers in a suitable manner. Thus, it can be published as is now.